# Promotive and protective effects of community-related positive childhood experiences on adult health outcomes in the context of adverse childhood experiences: a nationwide cross-sectional survey in Japan

Haruyo Mitani [1], Naoki Kondo [2], Airi Amemiya [2], Takahiro Tabuchi [3,4]

For numbered affiliations see end of article.

**Correspondence to**
Dr Haruyo Mitani;
mitani.haruyo.hus@osaka-u.ac.jp

## ABSTRACT

**Objective** Although adverse childhood experiences (ACEs) are associated with poor health in adulthood, positive childhood experiences (PCEs) can reduce the risk of negative health outcomes. This study aimed to investigate whether PCEs in the community (CPCEs, ie, trusted adults other than parents, supportive friends, belongingness to school, or community traditions) would have an independent effect on better health outcomes and moderate the association between ACEs and adult illnesses.

**Design** Cross-sectional survey.

**Setting** Data were gathered from a nationwide, cross-sectional internet survey conducted in Japan in 2022.

**Participants** This study included 28 617 Japanese adults aged 18–82 years (51.1% female; mean age=48.1 years).

**Primary and secondary outcome measures** The associations among self-reported ACEs, CPCEs before the age of 18 years and current chronic diseases (eg, cancer and depression) were investigated using multivariable logistic regression models.

**Results** CPCEs were associated with lower odds of adult diseases (such as stroke, chronic obstructive pulmonary disease (COPD), chronic pain, depression, suicidal ideation and severe psychological distress) after adjusting for ACEs. More CPCEs weakened the association between ACEs and adult diseases. Specifically, among those with ACEs, ≥3 CPCEs (vs 0–2 CPCEs) lowered the adjusted prevalence by ≥50% for stroke (2.4% to 1.2%), COPD (2.2% to 0.7%) and severe psychological distress (16.4% to 7.4%).

**Conclusion** CPCEs could reduce ACE-related risk of poor physical and mental health in later life. Early-life interventions that enhance PCEs in schools and/or neighbourhoods are recommended.

## INTRODUCTION

Over the last quarter century, a growing body of research has demonstrated the harmful effects of adverse childhood experiences (ACEs), such as maltreatment and exposure to household dysfunction, on long-term physical and mental health.[1–6] Studies have emphasised a dose–response relationship between an individual's ACE score and their risk of having poor health; that is, the greater the exposure to ACEs, the greater the risk of physical and mental illness in later life. Chronic childhood stress exposure can lead to dysregulation of stress response systems, including the hypothalamic–pituitary–adrenocortical axis and the sympathetic–adrenomedullary system, resulting in permanent changes in multiple organ systems, particularly the brain.[7] Epigenetic modifications of DNA could play a role in these changes, and the biological disruptions of brain circuits, other organs and metabolic systems during sensitive development are thought to lead to an increased risk of various chronic diseases in adulthood.[8]

Meanwhile, recent evidence shows a potential role of positive childhood experiences (PCEs). For example, continual early assistance from a reliable adult,[9] an adult who felt

## STRENGTHS AND LIMITATIONS OF THIS STUDY

⇒ This study used a large and representative sample from a nationwide survey in Japan.
⇒ Multivariable logistic regression models were used to adjust for possible confounders and to evaluate the associations among adverse childhood experiences (ACEs), positive childhood experiences in the community (CPCEs) and adult diseases.
⇒ Due to the self-reported and retrospective nature of the survey data, the possibility of recall bias cannot be eliminated.
⇒ A cross-sectional design limited our ability to determine causality.
⇒ The CPCEs measure used in this study did not include all CPCEs.

safe and protected as a child[10] or PCEs inside and outside the home (cumulative scale consisting of seven items such as 'able to talk to family about feelings' and 'felt supported by friends'),[11] moderates the association between ACEs and negative consequences, represented by poor mental health. In other words, PCEs have a moderating effect as a protective factor; that is, the harmful effects of ACEs can be smaller among individuals who report more PCEs than among those who report fewer PCEs.[9–15]

Resilience refers to the latent or apparent capacity to successfully respond to situations that threaten an individual's functioning, survival or development.[12] Based on the resilience theory,[12] the socioecological system suggests that multiple systems (eg, individuals, families and communities) complementarily influence youth development. Moreover, factors promoting resilient outcomes can be divided into individual (eg, cognition, coping style and self-esteem), family (eg, a safe and supportive relationship with a caregiver) and community factors (eg, positive relationships with friends and neighbours).[12–15] Notably, the concept of PCEs includes both family and community factors. Little is understood about the potential of PCEs in the community (CPCEs) for countering the negative lifetime effects of ACEs.

This study focused on the role of community-related PCEs (CPCEs) outside the home. Compared with those with fewer ACEs, children exposed to a higher number of ACEs are less likely to report having adults in their lives who made them feel safe or met their basic needs, highlighting a deficit in PCEs.[10] In other words, those with more ACEs are less likely to have experienced PCEs in the home. Hence, continuous family intervention by professionals to change how caregivers relate to children is desirable to enrich family PCEs (FPCEs).[16] However, many countries lack the infrastructure (staff, funds and legal structure) to make such professional intervention possible, and Japan is no exception.[17] In contrast, community interventions in schools and neighbourhoods can easily and inexpensively positively impact the development of children exposed to ACEs. Clarifying the role of CPCEs is essential for understanding the effectiveness of community interventions.

Therefore, this study explored the effect of CPCEs on the negative consequences of ACEs, focusing on four items of the PCE scale,[11] referenced in many studies, which are considered particularly present in the community (ie, trusted adults other than parents, supportive friends, belongingness to school and community traditions). Similar to the original ACE study,[1] this study used a broad range of physical and mental illnesses as primary outcomes. In Japan, while ACEs influence physical illnesses in old age[18] and mental disorders in adults,[19] the interplay between PCEs and ACEs on health remains unexplored. Stress is identified as a contributing factor to adult diseases.[20] ACEs are considered stressors originating within the childhood home, whereas CPCEs are viewed as external factors that mitigate the impact of these stressors.

This study explored the associations among ACEs, CPCEs and adult diseases. First, the relationship between ACEs and adult diseases was ascertained. Then, the resilience theory's three main models (the compensatory model, the protective factors model and the challenge model[12 21–23]) were examined.

The compensatory (promotive factors) model assumes that promotive/compensatory factors directly impact developmental outcomes independent of risk factors. For example, even after controlling for ACEs, PCEs have been reported to predict less post-traumatic syndrome disorder symptoms and stress in pregnant women[24] and less depression, stress and sleep difficulties in adults.[25] While ACEs could still have a significant effect on the outcome, CPCEs themselves are thought to enhance health in adulthood.

The protective factors model assumes that protective factors mitigate the relationship between risk factors and bad outcomes. Protective factors do not act independently but can interact with risk factors. Previous studies have shown that adults who protect or meet basic needs[10] and high PCE scores[11] were associated with less poor mental health, especially among those with ACEs. It has also been found that in those with ≥4 ACEs, having all childhood community resilience assets such as given opportunity, supportive friends and role model (vs none) lowered the prevalence of self-rated poor childhood health from 59.8% to 21.3%.[26] In situations where more CPCEs are reported, the negative impact of ACEs on adult health is likely to be smaller.

The challenge model assumes that the modest levels of risk factors protect against additional exposures that could render people prone to unfavourable consequences. Moderate levels of ACEs, particularly when accompanied by more protective factors, may promote improved adult health, akin to inoculation.[22] Some studies did not support this model,[21] but it has also been reported that PCE score had greater positive effects on adult health (such as body mass index and sleep difficulties) among those with <4 ACEs than among those with ≥4 ACEs.[25] Therefore, it can be speculated whether moderate ACEs can lower the likelihood of adult illness among individuals reporting more CPCEs.

Thus, this study tested three hypotheses: CPCEs directly affect adult health (H1), the association between ACEs and adult diseases is weaker among people with more CPCEs (H2) and ACEs are negatively associated with adult disease among people with more CPCEs (H3).

## METHODS
### Sample
The Japan COVID-19 and Society Internet Survey (JACSIS)[27] was conducted in 2022 by a major internet research agency (Rakuten Insight) with around 2.2 million registered qualified panellists. Of the panellists, 224 389 males and females aged 15–79 were invited to participate in the survey based on a random sample stratified by sex,

age and all 47 prefectures. Participants were rewarded with 'E-Points' for online shopping. Informed consent was obtained from all participants before the survey. The survey began on 12 September 2022, and ended on 19 October 2022. The survey was terminated after the target of 32 000 participants was reached.

Participants who provided contradictory answers to the screening questions were excluded from the study. These included the following criteria: (1) invalid answers to the screening question (participants were asked to choose the second-last option, but they chose other options), (2) positive responses to all items on drug use, (3) positive answers for having all 20 chronic diseases and (4) a total household size of over 15 individuals. There were 3370 respondents who gave inconsistent answers, leaving 28 630 individuals. 13 participants aged <18 were also excluded since ACEs and PCEs are answered by recalling experiences up to age 18. Finally, 28 617 adults were included. There were no significant differences in the key variables (ACEs, PCEs and adult health outcomes) between 28 617 adults included in the analysis and 3370 adults not included in the study. Due to the nature of the online survey, there were no missing values.

### Patient and public involvement
Neither patients nor the public were involved in this study.

### Measurements
#### Adverse childhood experiences
Questions on ACEs were derived from the ACEs scale for the Japanese context.[28] Although this study was based on the original ACE study,[1] ACE items included parental death, except the incarceration of household members, to fit the Japanese context. Respondents were questioned whether any of the following adversities had occurred in their households by age 18: (1) death of either parent; (2) parental divorce or separation; (3) mental illness of either parent; (4) substance abuse in the household (eg, parental addiction to alcohol and gambling); (5) mother treated violently (the father was violent with the mother); (6) emotional abuse (my parent said hurtful things or insulted me); (7) physical abuse (my parent beat and hurt me); (8) sexual abuse (an adult sexually touched me); (9) emotional neglect (I always felt suffocated because my parent did not respect my opinion) and (10) physical neglect (I did not receive necessary care, such as eating and dressing). The response categories 'yes' and 'no' were coded as dummy variables, and these 10 items were summed into ACE scores (0–10). This score was classified into two categories (0 vs ≥1) because of their effectiveness in determining poor health, following those of previous studies[18 29] and the clarity of the semantic difference between 'without ACEs' and 'with ACEs'.

#### Positive childhood experiences
PCEs were assessed using seven items adapted from the PCE scale.[11] Participants answered whether they had experienced any of the following by the time they were 18

years old: (1) felt able to talk to their family about feelings; (2) felt their family stood by them during difficult times; (3) felt safe and protected by an adult in their home; (4) had at least two non-parental adults who took genuine interest in them; (5) felt supported by friends; (6) felt a sense of belonging in middle and high school and (7) enjoyed participating in community traditions. Although these questions were asked in Japanese, and some were modified from the original questions (the addition of 'middle' in item 6), these seven items can be considered almost identical to the PCEs items used by Bethell *et al.*[11] Of these items, the first three were positioned as components of FPCEs and the others as CPCEs (online supplemental table S1). The applicable items were summed to obtain the FPCE (0–3) and CPCE (0–4) scores. As the PCE scale,[11] principal component analyses revealed a single-component structure for each FPCE and CPCE scale. Cronbach's alpha was 0.81 and 0.75 for FPCEs and CPCEs, respectively (online supplemental table S2).

### Adult health outcomes
Participants answered whether they currently had the following diseases using a dichotomous response variable (yes/no): (1) diabetes, (2) ischaemic heart disease (angina pectoris or myocardial infarction), (3) stroke (cerebral infarction or haemorrhage), (4) chronic obstructive pulmonary disease (COPD), (5) cancer, (6) chronic pain (eg, back pain and headache lasting ≥3 months) and (7) depression. Additionally, (8) suicidal ideation was determined as a 'yes' response to the question, 'Have you had times when you have wanted to die in the last year?' Psychological distress was measured with the Japanese version[30] of the Kessler 6-item psychological distress scale,[31] assessing depressed mood and anxiety. The total score ranged from 0 to 24 on a 5-point scale (0–4). A score of ≥13 was considered indicative of (9) serious psychological distress.[32]

### Covariates
The following background variables reported by participants were included as covariates, in line with previous studies[1 33 34]: age (continuous by year), sex (0 'male' or 1 'female') and educational attainment (three dummy variables for 'less than high school', 'high school' and 'post-secondary and graduate').

### Statistical analysis
Demographic characteristics and prevalence rates of all health outcomes in the sample were examined. For H1, multivariable logistic regression analyses were conducted to consider the direct effect of CPCEs on adult diseases after adjusting for covariates, ACEs and FPCEs. For H2 and H3, multiple logistic regression analyses, stratified by CPCE exposure levels, were performed to examine whether CPCEs moderated the link between ACEs and adult diseases, adjusting for covariates and FPCEs. Interactive effects and marginal predicted probabilities were estimated to assess the statistical significance of the

moderating effects of CPCEs on the association between ACEs and adult diseases.[35] The extent to which CPCEs reduced the risk of each disease was also estimated.

The survey data were weighted to represent the Japanese population. Considering the possibility of multicollinearity, the variance inflation factor (VIF) for each explanatory variable was checked to ensure that it did not exceed 4.0.[36] Stata V.17.0 MP (StataCorp) was used for all analyses.

## RESULTS
### Participant characteristics
Table 1 shows that chronic pain and COPD were the most prevalent (18.6%) and least prevalent (1.4%) of the nine health outcomes, respectively. Approximately 39.8% of the participants had experienced ACEs. Furthermore, 52.8% and 56.0% of participants had experienced one or more FPCEs and CPCEs, respectively. When ACEs were divided into four exposure levels,[9 11] the more exposure to ACEs, the higher the prevalence of adult diseases, the lower the exposure rates to the two types of PCEs, and the higher the proportion of females and those with lower education levels (online supplemental table S3). Finally, dose–response relationships between ACE exposure levels and all adult diseases were observed (online supplemental table S4).

### Health outcomes according to ACE, FPCE and CPCE exposure
Table 2 shows the adjusted ORs (AORs) for the poor health associated with CPCEs. In all models, VIF values did not exceed the criterion value (4.0). CPCEs had significant negative direct effects on six adult diseases: stroke, COPD, chronic pain, depression, suicidal ideation and severe psychological distress, after adjusting for covariates, ACEs and FPCEs. This result supported H1, which states that CPCEs directly affect adult health, derived from the compensatory model. For example, the adjusted odds of stroke were 15% lower for each additional CPCE. The same trend was found when CPCEs were treated as categorical variables (online supplemental table S5). Overall, CPCEs were particularly associated with lower odds of mental illness.

### Health outcomes of ACEs across CPCE exposure levels
As shown in figure 1, the AORs declined as PCE exposure levels rose; the cells became greener to the right. Particularly, the AORs tended to be smaller when the CPCE scores were ≥3 than when they were 0–2. This value of 3 CPCEs is also consistent with the Aiken and West[37] method, which recommends that the cut-off value is close to the mean plus one standard deviation (mean 1.25+SD 1.4=2.65). Therefore, the interaction effects of ACEs and CPCEs (≥3 vs 0–2) on all nine outcomes were examined (online supplemental table S6). Although the rightmost column of figure 1 shows that the regression coefficients on the interaction terms between any ACE and ≥3 CPCEs were significant at the 10% level for seven diseases, it is

**Table 1** Demographic characteristics and prevalence of health outcomes, ACEs and PCEs among participants (N=28 617)

|  | N | W%/M (SD) |
|---|---|---|
| **Health outcomes** | | |
| Diabetes | 1945 | 7.7 |
| Ischaemic heart disease | 686 | 3.0 |
| Stroke | 433 | 1.7 |
| COPD | 381 | 1.4 |
| Cancer | 705 | 2.6 |
| Chronic pain | 5062 | 18.6 |
| Depression | 1381 | 4.7 |
| Suicidal ideation | 3909 | 14.4 |
| Severe psychological distress | 2881 | 10.4 |
| **ACEs (0–11)** | 28 617 | 0.81 (1.4) |
| 0 ACEs | 17 889 | 60.2 |
| ≥1 ACEs | 10 728 | 39.8 |
| **FPCEs (0–3)** | 28 617 | 1.14 (1.2) |
| 0 FPCE | 12 003 | 47.2 |
| 1 FPCEs | 4133 | 14.0 |
| 2 FPCEs | 5025 | 16.2 |
| 3 FPCEs | 7456 | 22.7 |
| **CPCEs (0–4)** | 28 617 | 1.25 (1.4) |
| 0 CPCE | 11 051 | 44.0 |
| 1 CPCEs | 5543 | 19.4 |
| 2 CPCEs | 4502 | 14.7 |
| 3 CPCEs | 3922 | 11.5 |
| 4 CPCEs | 3599 | 10.5 |
| **Age (18–82 years)** | 28 617 | 48.1 (17.2) |
| **Sex** | | |
| Male | 13 993 | 48.9 |
| Female | 14 624 | 51.1 |
| **Educational attainment** | | |
| Less than HS | 493 | 4.4 |
| HS graduate | 10 516 | 60.3 |
| Postsecondary and graduate | 17 397 | 33.8 |

ACEs, adverse childhood experiences; COPD, chronic obstructive pulmonary disease; CPCEs, community positive childhood experiences; FPCEs, family positive childhood experiences; HS, high school; W, weighted.

a better way to examine the predicted probability themselves and perform the test of the equality of marginal effects to determine whether significant interaction effects exist.[35]

Figure 2 shows the predicted values for each disease obtained from the regression equations. The results of the marginal effect difference test (the red values of figure 2 and online supplemental table S7) showed significant differences in the marginal effects (the amount of

**Table 2** Adjusted ORs of health outcomes by ACEs (≥1 vs 0), FPCEs and CPCEs

| | ACEs | | FPCEs | | CPCEs | |
|---|---|---|---|---|---|---|
| | AOR* | (95% CI) | AOR† | (95% CI) | AOR‡ | (95% CI) |
| Diabetes | **1.31** | (1.18 to 1.44) | **0.93** | (0.89 to 0.99) | 0.97 | (0.93 to 1.02) |
| Ischaemic heart disease | **1.53** | (1.31 to 1.80) | **0.89** | (0.82 to 0.97) | 0.93 | (0.87 to 1.01) |
| Stroke | **1.96** | (1.61 to 2.39) | **0.83** | (0.75 to 0.93) | **0.85** | (0.77 to 0.93) |
| COPD | **2.00** | (1.62 to 2.47) | **0.77** | (0.69 to 0.86) | **0.89** | (0.81 to 0.98) |
| Cancer | **1.62** | (1.39 to 1.90) | **0.89** | (0.82 to 0.97) | 0.95 | (0.89 to 1.02) |
| Chronic pain | **1.71** | (1.60 to 1.82) | 0.98 | (0.95 to 1.02) | **0.91** | (0.89 to 0.94) |
| Depression | **2.86** | (2.55 to 3.22) | **0.89** | (0.83 to 0.95) | **0.81** | (0.76 to 0.86) |
| Suicidal ideation | **2.12** | (1.97 to 2.28) | **0.84** | (0.81 to 0.88) | **0.77** | (0.74 to 0.80) |
| Severe psychological distress | **2.29** | (2.10 to 2.48) | **0.86** | (0.81 to 0.90) | **0.72** | (0.69 to 0.75) |

N=28617; significance level of data in bold font <0.05.
*Adjusted for age, sex, educational attainment, FPCEs and CPCEs.
†Adjusted for age, sex, educational attainment, ACEs and CPCEs.
‡Adjusted for age, sex, educational attainment, ACEs and FPCEs.
ACEs, adverse childhood experiences; AOR, adjusted OR; COPD, chronic obstructive pulmonary disease; CPCEs, community positive childhood experiences; FPCEs, family positive childhood experiences.

change in predicted probability when ACEs exceed ≥1 from 0) between those with 0–2 CPCEs and those with ≥3 CPCEs for eight diseases except diabetes. CPCEs moderated the association between ACEs and eight adult diseases. This result supported H2, which states that the association between ACEs and adult diseases is weaker among people with more CPCEs, derived from the

protective factors model. Specifically, in ischaemic heart disease, the difference between the marginal effect of 0–2 CPCEs (0.034–0.21=0.013) and one of ≥3 CPCEs (0.017–0.019=−0.001) was 0.015, significant at the 1% level. According to each marginal effect (online supplemental table S7), the marginal effect for CPCEs≥3 was negative for ischaemic heart disease (−0.001, 95% CI −0.008

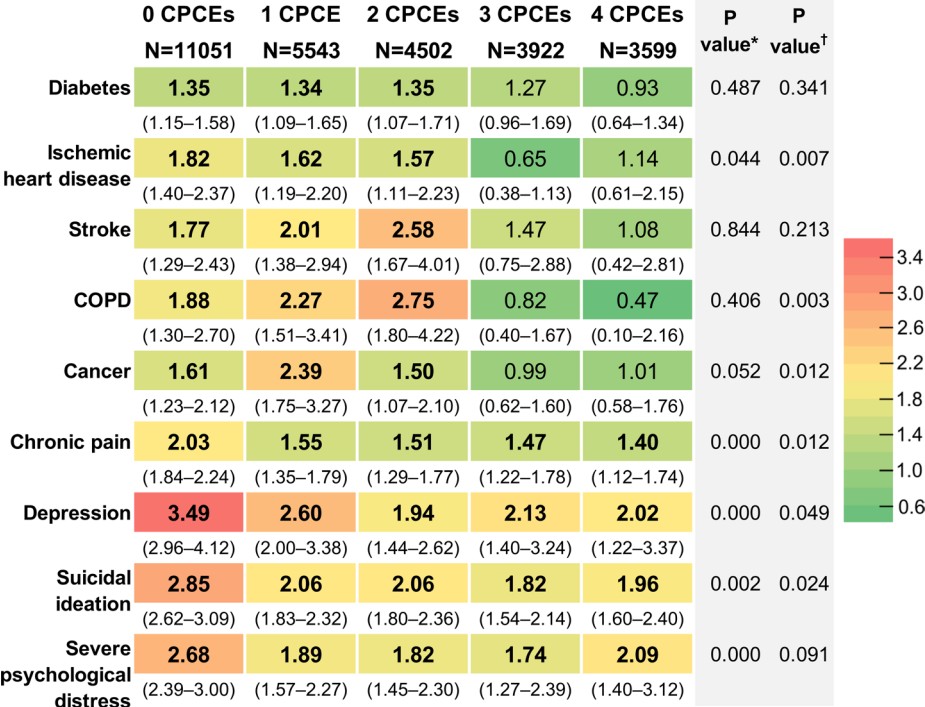

**Figure 1** A heatmap of the adjusted ORs with 95% CIs for the health outcomes of ACEs (≥1 vs 0) across CPCE exposure levels. All ORs are adjusted for age, sex, educational attainment and FPCEs. The significance level of data in bold font is <0.05. *P value for the interaction effect of ACEs (≥1 vs 0)×CPCEs (cumulative variable). †P value for interaction effect of ACEs (≥1 vs 0)×CPCEs (≥3+ vs 0–2). ACEs, adverse childhood experiences; COPD, chronic obstructive pulmonary disease; CPCEs, community positive childhood experiences; FPCEs, family positive childhood experiences.

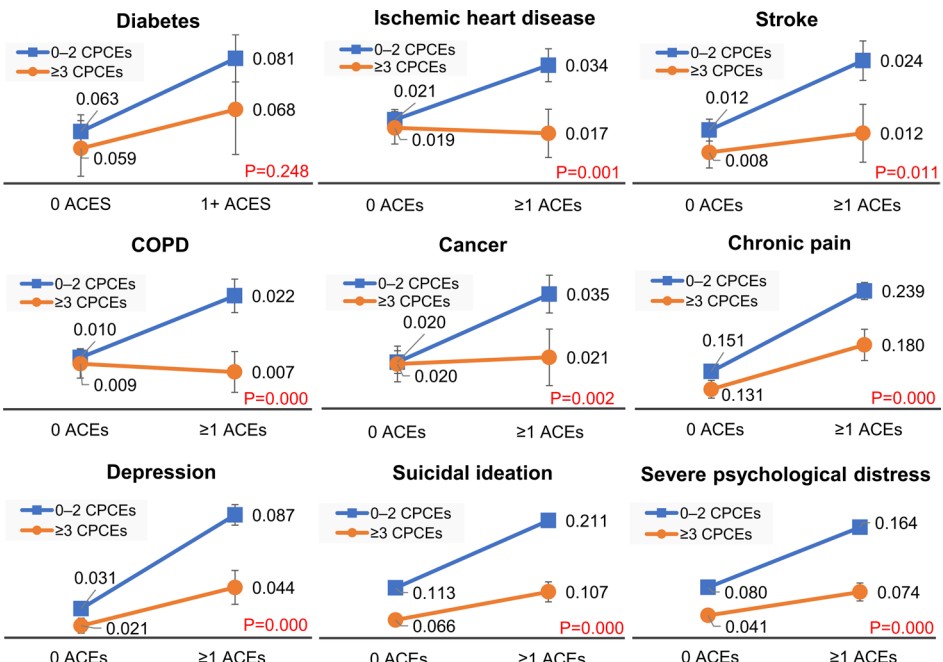

**Figure 2** Adjusted predictions with 95% CIs for the health outcomes of ACEs (≥1 vs 0)×CPCEs (≥3 vs 0–2). Substitutions were performed on the multiple logistic regression models presented in online supplemental table S6. P value for significant differences of marginal effects between 0–2 and ≥3 CPCEs. ACEs, adverse childhood experiences; COPD, chronic obstructive pulmonary disease; CPCEs, community positive childhood experiences.

to 0.006) and COPD (−0.002, 95% CI −0.006 to 0.003). A negative association between ACEs and adult disease in the context of reporting more CPCEs was suggested, but H3 was not statistically supported, as suggested by the challenge model.

From figure 2, the differences in predicted morbidity by CPCEs among those with ACEs can also be captured as relative risk reduction. For example, the predicted prevalence of ischaemic heart disease did not differ according to CPCE exposure among those without ACEs; however, among those with ACEs, it differed from 3.4% in those with 0–2 CPCEs to 1.7% in those with ≥3 CPCEs (relative risk: 1.725/3.400×100=50.794%, relative risk reduction: 100.0−50.8=49.2%). The adjusted prevalence was reduced by almost ≥50% for ischaemic heart disease (49.2%), stroke (51.9%), COPD (67.1%), depression (49.1%) and severe psychological distress (54.9%).

## DISCUSSION

This study examined the associations among ACEs, CPCEs and adult diseases using a population-based sample from Japan. Specifically, ACEs were associated with increased odds of adult diseases while CPCEs directly affected adult illnesses as a promotive factor and mitigated the adverse health effects of ACEs as a protective factor. The protective effect of CPCEs was particularly pronounced in stroke, COPD and severe psychological distress, with associated relative risk reductions of 50% or more by three or more CPCEs in those with ACEs.

Consistent with earlier research,[1 4 6] the results showed a dose–response connection between ACEs and all adult

diseases evaluated. Although an association between ACEs and diseases during old age among Japanese individuals was previously reported,[18] this appeared robust across a wide age range (18–82 years). The ACE items are based on the original ACE study.[1] Although this measure does not include the variety of traumatic experiences that participants may have, it is noteworthy that a dose–response relationship with adult health was found, as in many studies.

H1, based on the compensatory model, was supported for many of the diseases assessed in this study. CPCEs, independent of ACEs and FPCEs, were directly associated with lower odds of adult-onset illnesses (ie, stroke, COPD, chronic pain, depression, suicidal ideation and severe psychological distress). While previous studies have found direct effects of PCEs primarily on mental health,[11 25 34] this study adds a novel finding that CPCEs independently function as promotive factors for physical and mental health.

More importantly, this study found analytical support for H2 based on the protective model. CPCEs moderated the relationships between ACEs and adult diseases (ie, ischaemic heart disease, stroke, COPD, cancer, chronic pain, depression, suicidal ideation and severe psychological distress), even after adjusting for FPCEs. Although the possible protective function of PCEs has been noted,[9–11] the buffering effect may be due to family factors. Notably, this study observed that CPCEs can independently play a protective role. Furthermore, extending research focusing on community assets moderating the association between ACEs and poor child health,[26] this

study suggests that the moderating effects of CPCEs may spread to adulthood, indicating that the protective effect may be stronger in cumulative situations where three or more CPCEs are aligned. The associated relative risk reductions of physical illness (ie, ischaemic heart disease, stroke and COPD) and mental illness (ie, depression and severe psychological distress) in ACE survivors by almost more than half due to ≥3 CPCEs would be a non-negligible value. Whether this cut-off point is appropriate requires follow-up testing; however, the harm caused by ACEs may be mitigated in multiple relationships.

Moreover, the analysis yielded the finding that, for several diseases (ie, ischaemic heart disease and COPD), those with ACEs had lower prevalence rates than those without ACEs in situations reporting more CPCEs (figure 2). Although the negative marginal effects of ACEs on these illnesses were not statistically significant, as the challenge model[22] of the resilience theory implies, moderate ACEs promote health when sufficient protective factors exist; that is, they act as immunisation. Future studies should examine the generalisability of this phenomenon.

Why might CPCEs contribute to developing children and youth into healthier adults? Prior research argues that supportive adults other than parents may provide opportunities for children to elude life stressors, establish positive experiences in social connections and feel a sense of having a place and social association.[38] Additionally, supportive close friendships promote resilience by developing constructive coping styles, encouraging efforts, providing supportive peer networks and reducing inappropriate coping.[39] In interactions with non-family members at school and in the neighbourhood, children exposed to ACEs may gain physical refuge from toxic stress and develop psychological and instrumental coping strategies. The role of CPCEs as a promoting and protective factor found in this study may have been discovered only in the context of Japan, where many local communities have self-governing organisations and traditional events, and violence among residents and students is relatively rare. Future studies should examine whether the findings are generalisable to other countries besides Japan.

### Strengths and limitations

To the best of our knowledge, this is the first study to suggest that CPCEs could be promotive and protective factors of health system resilience. A strength of this study is the use of a large and representative sample. Moreover, various indicators elucidated the complex relationships among ACEs, PCEs and adult diseases.

However, this study had some limitations. First, the data were self-reported and retrospective, which could have been influenced by recall bias. Individuals with a disease may be more likely to recall ACEs than healthy individuals. However, asking participants to reflect on their growing environment from a balanced perspective by asking about PCEs may have reduced this recall bias. Second, because of the study's cross-sectional nature, the possibility of reverse causation between ACEs/PCEs and adult illnesses cannot be eliminated. This study could not allow for the determination of causality. Third, the CPCEs measure used in this study did not include all CPCEs. Other types of CPCEs, such as participation in non-traditional activities and self-realisation experiences, should be included in developing the CPCEs scale. Fourth, confounding variables (including psychological traits like optimism and hometown regional features, which may be linked to adult diseases, ACEs and CPCEs) might have gone unnoticed. Fifth, the extent of the positive effects of the CPCEs differed for each health outcome. Particularly, the role of CPCEs in mental illness was found to be robust; however, the validity of these effects should be carefully investigated by including other health outcomes. Sixth, this study could not address a variety of traumatic experiences (eg, homelessness, bullying victimisation and childhood illness) other than typical ACEs.

### Policy, practice and research implications

The findings suggest that community-based interventions can be effective in helping children who have experienced adversity become resilient adults. Public health, social welfare and educational practices that enforce CPCEs might as well be promoted. Additionally, the content of specific community-level approaches depends on social and local contexts. Such interventions are desirable to incorporate the main components of the healthy outcomes from positive experiences framework: (1) supportive and nurturing relationships, (2) safe and stable environments, (3) opportunities for constructive social involvement and connection and (4) social and emotional capacities.[40] For example, The National Resource Center at Tufts Medical Center[41] supports the above framework, providing tools and training to help healthcare providers, social service providers, educators and members of community-based organisations acquire essential knowledge and skills to integrate these elements into their work.

In Japan, school counsellors[42] and school social workers[43] in public elementary and junior high schools are expected to become resilience resources under compulsory education. Additionally, teachers might be a good way to get training in PCEs and improved working environments to build trusting relationships with their students. In local communities, traditional events such as festivals, nature experiences, multigenerational exchange programmes, and the recently popularised 'Kodomo Shokudo' (cafeterias for children)[44] could also contribute to the children's resilience. Further research and practice are expected to explore effective approaches to promoting CPCEs, including historical and new engagements.

### Conclusion

This study showed that CPCEs positively impacted adult health in the general population regardless of ACE exposure and mitigated the adverse consequences. The study demonstrates that CPCEs may

contribute to the lifelong health status of ACE survivors. Therefore, further research is needed to investigate the impact mechanisms of CPCEs and improve the scale's accuracy while capturing diverse childhood adversities, as well as interventions to strengthen supportive communities.

**Author affiliations**
[1]Graduate School of Human Sciences, Osaka University, Suita, Osaka, Japan
[2]Department of Social Epidemiology, Graduate School of Medicine and School of Public Health, Kyoto University, Kyoto, Japan
[3]Cancer Control Center, Osaka International Cancer Institute, Osaka, Japan
[4]Division of Epidemiology, School of Public Health, Graduate School of Medicine, Tohoku University, Sendai, Miyagi, Japan

**Contributors** HM and AA conceptualised the study. HM conducted the analysis and drafted the first manuscript. NK and AA contributed intellectual content to the study design and interpretation of the findings. TT collected data, managed the project and obtained funding. All authors critically reviewed the manuscript and approved the final manuscript. HM is the guarantor for this study.

**Funding** This study (JACSIS2022) was supported by the Japan Society for the Promotion of Science (JSPS) KAKENHI Grants (21H04856, 20K10467, 20K19633, 20K13721), the JST Grant (JPMJPF2017), the Health Labor Sciences Research Grant (21HA2016), the grant for 2021–2022 Strategic Research Promotion (SK202116) of Yokohama City University, and the research program on 'Using Health Metrics to Monitor and Evaluate the Impact of Health Policies,' conducted at the Tokyo Foundation for Policy Research. HM was also supported by the JSPS KAKENHI Grant (21K13454) and the Ministry of Education, Culture, Sports, Science and Technology (MEXT) KAKENHI Grant (20H05805).

**Competing interests** None declared.

**Patient and public involvement** Patients and/or the public were not involved in the design, or conduct, or reporting, or dissemination plans of this research.

**Patient consent for publication** Not applicable.

**Ethics approval** This study involves human participants and was authorised by the Research Ethics Committee of the Osaka International Cancer Institute (approved on 14 June 2022; approval No. 20084-8). Participants gave informed consent to participate in the study before taking part.

**Provenance and peer review** Not commissioned; externally peer reviewed.

**Data availability statement** Data are available on reasonable request. All deidentified data from participants reported in this study are available to interested researchers who submit data-sharing requests to the last author with a summary of the secondary analysis plan.

**ORCID iDs**
Haruyo Mitani http://orcid.org/0000-0001-8882-5098
Naoki Kondo http://orcid.org/0000-0002-6425-6844
Airi Amemiya http://orcid.org/0000-0001-5319-9541
Takahiro Tabuchi http://orcid.org/0000-0002-1050-3125

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
