## [Reviewer comments · BMJ Open]

ARTICLE DETAILS

TITLE (PROVISIONAL)	Promotive and protective effects of community-related positive childhood experiences on adult health outcomes in the context of adverse childhood experiences: a nationwide cross-sectional survey in Japan
AUTHORS	Mitani, Haruyo; Kondo, Naoki; Amemiya, Airi; Tabuchi, Takahiro

VERSION 1 – REVIEW

REVIEWER	Mueller , Kyle Curtis Research Analyst at Harris County Juvenile Probation Department
REVIEW RETURNED	12-Dec-2023

GENERAL COMMENTS	Below I provide several comments that may strengthen this piece before publication. This piece explores the association between ACEs, PCEs, and acquiring disease during adulthood among a cross-sectional internet survey sample of Japanese participants aged 18-79 While interesting, and certainly important to research in this area, there are some shortcomings present in the piece as currently written. I enjoyed reading this manuscript and commend the authors on thinking seriously about positive life experiences, something that has been too often neglected in empirical articles in general in favor of a deficit approach. I also appreciated that the authors tried to stay true to the original ACE operationalization. While I do have some concerns about ACEs (mentioned below), the trend to add experiences and call them ACEs is becoming increasingly problematic in this space and I am glad to see that authors are not muddying the waters. I would also like to see a reminder that a cumulative ACE score is not, in and of itself, indicative of much. Trauma-informed treatment is meaningful and important without counting to ten. And some ACEs, such as divorce, may improve youth's lives, by separating them from abusive parents, for instance. Alternatively, the mechanism by which it affects youth may be different than the separation of a parent and more in line with the known effects of a drop in socio-economic status. It is also worth noting that many traumatic events are not included here, such as major physical and/or mental health concerns for the respondent, homelessness (or near homelessness), and or other forms of non-family-related traumatic victimization. ACEs as a tool are useful but by no means the only set of adverse childhood events that matter. The limitations, conclusions, and discussion need to address this more fully.
--

Given the importance of the Compensatory Model, the Protective Factor Model, and the Challenge Model, each of these need to be discussed in greater detail, and expectations stemming from them (concerning the association between ACEs/PCEs and adult illnesses) should be made more explicit. While they are each mentioned in a sentence or two, each requires some elaboration as they are central to the research questions and hypotheses that stem from them. Also, potentially the researchers should review the provided article citation below and test all models of resiliency theory instead of just the compensatory model and the protective model.

- Mueller, K. C., & Carey, M. T. (2023). How Positive and Negative Childhood Experiences Interact with Resiliency Theory and the General Theory of Crime in Juvenile Probationers. *Youth Violence and Juvenile Justice*, 21(2), 130-148.

There is also room for theoretical expansion about the association between PCEs and adult illnesses. Some review of factors associated with adult illnesses and their relationship to the measures used within ACEs and PCEs is warranted.

The logistic regression models and the research questions stemming from this article need to be better linked to testing the Compensatory model and Protective factors model. It is unclear how adding PCEs to a model assesses the potential of “neutralization,” as that is more akin to a mediation relationship, although the hypotheses offered later read more like individual main effects. There is also the examination of whether PCEs moderate the relationship between ACEs and adult illnesses. The researchers should include in the statistical analysis section the missing data analysis provided in more detail below and the VIF values of the variables included in the analyses.

Notably, one of the main research hypotheses involves moderating effects. Along those lines, researchers have recently pointed out, that nonlinear dependent variables present a special case for testing and interpreting interactions (Mize, 2019). The current editors of the *American Sociological Review* have recently advised as follows: “The case is closed: don’t use the coefficient of the interaction term to conclude the statistical interaction in categorical models such as logit, probit, Poisson, and so on” (Mustillo et al. 2018, p. 1282). Instead, the authors should test the second derivatives of the predicted probabilities of juvenile court outcomes across the interacted variables (see Mize, 2019). As such, it is suggested that the authors re-do all analyses using interactive effects and marginal predicted probabilities to assess their statistical significance. Also, on page, 15 “Perhaps, as the Challenge model of the resilience theory implies, moderate ACES promote health when sufficient protective factors exist; that is, they act as immunization. “This sentence should be addressed utilizing the method above before stating this statement.

I would also like to see a reminder that a cumulative ACE score is not, in and of itself, indicative of much. Trauma-informed treatment is meaningful and important without counting to ten. And some ACEs, such as divorce, may improve youth's lives, by separating them from abusive parents, for instance. Alternatively, the mechanism by which it affects youth may be different than the separation of a parent and more in line with the known effects of a

	drop in socio-economic status. It is also worth noting that many traumatic events are not included here, such as major physical and/or mental health concerns for the respondent, homelessness (or near homelessness), and or other forms of non-family-related traumatic victimization. ACEs as a tool are useful but by no means the only set of adverse childhood events that matter. The limitations, conclusions, and discussion need to address this more fully. The researchers should show that the participants included in the analyses are missing at random rather than missing systematically in which multiple imputations would be needed to be utilized. Specifically, on page 6 out of 28,630 individuals, why were only 28,254 included in the study? Why did the researchers drop 376 observations? Why did the researchers choose the cut-points for the ACE and split PCE measures this should be justified within the text. Why would you not use a threshold of PCEs and ACES utilizing the method determined by following Aiken and West's (1991) suggestion for determining a cut point, which involves a value near the mean plus one standard deviation.  • Aiken, L. S. (1991). Multiple regression: Testing and interpreting interactions. Sage Publications google schola, 2, 513-531. Limitations: The researchers should also include the cross-sectional nature of the data, removing the ability to establish causality more specifically adding a citation or citations would be useful. The aforementioned theoretical and methodological concerns must be addressed, even if not all of them can be completely fixed in this one study.
--	--

REVIEWER	Gissandaner , Tre D. Columbia University Irving Medical Center
REVIEW RETURNED	21-Dec-2023

GENERAL COMMENTS	Thank you for the opportunity to review this manuscript. This manuscript examined the promotive and protective effects of community-related positive childhood experiences on adult health outcomes in the context of adverse childhood experiences. This examination is relevant and utilizes a large representative sample. The article is overall well written and concise. There are some mostly minor concerns related to study rationale, as well as improving clarity of other aspects of the manuscript. Specific comments and suggestions are detailed below, organized by manuscript sections. Introduction 1. It would be helpful to the reader to briefly define what a dose-response relationship means. Then a follow-up sentence explaining the implication of this on physical and mental health would provide better justification for the last sentence of this paragraph.
--

	2. Second paragraph. Again, defining or clarifying what the moderating effect of PCEs means for the reader would be helpful. 3. Paragraph 4. What is meant by “children exposed to ACEs often have ACE in their relationship with caregivers, not PCE”? 4. It would be more appropriate to say that compensatory factors directly impact an outcome independent of the risk factor rather than “neutralize” it. A risk factor could still have a significant effect on the outcome. Methods 1. Please include how covariate information was obtained and how they were coded for the analyses. 2. How was relative risk derived? 3. Was there missing data? Please include how missing data was handled? Results 1. Please avoid using causal language for this cross-sectional study. Instead of saying “increased,” “decreased,” or “reduced” say something like “was associated with increased odds” or “was associated with lower odds.” Discussion 1. Same as above for the discussion section. Please be careful with the causal language. 2. The study’s generalizability outside of Japan should be discussed. 3. In the implications section, language stating “should,” “need,” etc. should be softened. 4. Could there be additional confounding variables that might impact the internal validity of this study? This needs to be acknowledged.
--	--

VERSION 1 – AUTHOR RESPONSE

Reviewer #1
 Dr. Kyle Curtis Mueller
 Research Analyst at Harris County Juvenile Probation Department

Dear Dr. Kyle Curtis Mueller,

Thank you for your valuable comments and suggestions. Below, we have provided responses to your comments.

Responses to comments of Reviewer #1:

Comment 1: This piece explores the association between ACEs, PCEs, and acquiring disease during adulthood among a cross-sectional internet survey sample of Japanese participants aged 18-79 While

interesting, and certainly important to research in this area, there are some shortcomings present in the piece as currently written. I enjoyed reading this manuscript and commend the authors on thinking seriously about positive life experiences, something that has been too often neglected in empirical articles in general in favor of a deficit approach. I also appreciated that the authors tried to stay true to the original ACE operationalization. While I do have some concerns about ACEs (mentioned below), the trend to add experiences and call them ACEs is becoming increasingly problematic in this space and I am glad to see that authors are not muddying the waters.

Response 1: We appreciate all of your encouraging comments.

Comment 2: I would also like to see a reminder that a cumulative ACE score is not, in and of itself, indicative of much. Trauma-informed treatment is meaningful and important without counting to ten. And some ACEs, such as divorce, may improve youth's lives, by separating them from abusive parents, for instance. Alternatively, the mechanism by which it affects youth may be different than the separation of a parent and more in line with the known effects of a drop in socio-economic status. It is also worth noting that many traumatic events are not included here, such as major physical and/or mental health concerns for the respondent, homelessness (or near homelessness), and other forms of non-family-related traumatic victimization. ACEs as a tool are useful but by no means the only set of adverse childhood events that matter. The limitations, conclusions, and discussion need to address this more fully.

Response 2: We agree with your opinion. We have added the limitation of not having addressed a variety of traumatic experiences (e.g., homelessness, bullying victimization, childhood illness, etc.) other than typical ACEs and the need to capture diverse childhood adversities in discussion (page 15, lines 12-14), limitations (page 18, lines 3-5), and conclusion (page 19, line 10).

Comment 3: Given the importance of the Compensatory Model, the Protective Factor Model, and the Challenge Model, each of these need to be discussed in greater detail, and expectations stemming from them (concerning the association between ACEs/PCEs and adult illnesses) should be made more explicit. While they are each mentioned in a sentence or two, each requires some elaboration as they are central to the research questions and hypotheses that stem from them. Also, potentially the researchers should review the provided article citation below and test all models of resiliency theory instead of just the compensatory model and the protective model.

• Mueller, K. C., & Carey, M. T. (2023). How Positive and Negative Childhood Experiences Interact with Resiliency Theory and the General Theory of Crime in Juvenile Probationers. *Youth Violence and Juvenile Justice*, 21(2), 130-148.

Response 3: We agree with you. Thank you for providing the very helpful article. In accordance with this article, we have added more detailed descriptions of the three models of resiliency theory (page 6, lines 1-13). Also, a third hypothesis (H3) for the challenge model was set up to test adding H1 for the compensatory model and H2 for the protective factors model (page 6, lines 14-16).

Comment 4: There is also room for theoretical expansion about the association between PCEs and adult illnesses. Some review of factors associated with adult illnesses and their relationship to the measures used within ACEs and PCEs is warranted.

Response 4: We agree with you. We have discussed that stress is one of the causes associated with adult illnesses and that ACEs would be positioned as stressors within the childhood home and CPCEs as factors outside the home that soften the impact of those stressors (page 5, lines 18-20).

Comment 5: The logistic regression models and the research questions stemming from this article need to be better linked to testing the Compensatory model and Protective factors model. It is unclear how adding PCEs to a model assesses the potential of "neutralization," as that is more akin to a mediation relationship, although the hypotheses offered later read more like individual main effects.

There is also the examination of whether PCEs moderate the relationship between ACEs researchers should include in the statistical analysis section the missing data analysis provided in more detail below and the VIF values of the variables included in the analyses.

Response 5: We agree with you. Since the term “neutralization” was ambiguous, we have dropped it and expressed that “promotional/compensatory factors directly impact developmental outcomes independent of risk factors” (page 6, lines 1-2). Then, we have added a statement that there are no missing values (page 7, lines 16-17). All VIF values were checked (page 10, lines 17-19), and the description that they did not exceed the standard values was added in the text (page 12, line 4).

Comment 6: Notably, one of the main research hypotheses involves moderating effects. Along those lines, researchers have recently pointed out, that nonlinear dependent variables present a special case for testing and interpreting interactions (Mize, 2019). The current editors of the *American Sociological Review* have recently advised as follows: “The case is closed: don’t use the coefficient of the interaction term to conclude the statistical interaction in categorical models such as logit, probit, Poisson, and so on” (Mustillo et al. 2018, p. 1282). Instead, the authors should test the second derivatives of the predicted probabilities of juvenile court outcomes across the interacted variables (see Mize, 2019). As such, it is suggested that the authors re-do all analyses using interactive effects and marginal predicted probabilities to assess their statistical significance. Also, on page, 15 “Perhaps, as the Challenge model of the resilience theory implies, moderate ACES promote health when sufficient protective factors exist; that is, they act as immunization. “This sentence should be addressed utilizing the method above before stating this statement.

Response 6: Thank you for your useful suggestion. We have included that “it is a better way to examine the predicted probability themselves and perform the test of the equality of marginal effects to determine whether significant interaction effects exist” (page 13, line 10 to page 14, line 2), adding the results of the test to Figure 2 and new Supplementary Table S7. These results allowed us to draw a clear conclusion showing “significant differences in the marginal effects (the amount of change in predicted probability when ACEs exceed ≥ 1 from 0) between those with 0–2 CPCEs and those with ≥ 3 CPCEs for eight diseases except diabetes” (page 14, lines 5-7). By this examination of marginal effects, we could also have tested the H3 derived by the Challenge Model. We have shown that “a negative association between ACEs and adult disease in the context of an abundance of CPCEs was suggested, but H3 was not statistically supported” (page 14, lines 13-14).

Comment 7: The researchers should show that the participants included in the analyses are missing at random rather than missing systematically in which multiple imputations would be needed to be utilized. Specifically, on page 6 out of 28,630 individuals, why were only 28,254 included in the study? Why did the researchers drop 376 observations?

Response 7: These 376 individuals were either those aged <18 or ≥ 80 . Those aged <18 were excluded since ACEs and PCEs are answered by recalling experiences up to age 18 (page 7, lines 14-15). Participants aged >79 years ($n = 363$) were excluded from the analysis in the initial manuscript because they were beyond the population age range (15-79) assumed at the time of sampling (2019); however, we have reconsidered that there was no positive reason to exclude this sample. Including these 363 individuals did little to change the results of the analysis. Therefore, the analysis was reworked with 28,617 participants (page 7, line 16), excluding 13 minors from the 28,630, and all figures and tables were replaced. The figures in the text were slightly revised accordingly.

Comment 8: Why did the researchers choose the cut-points for the ACE and split PCE measures this should be justified within the text. Why would you not use a threshold of PCEs and ACES utilizing the method determined by following Aiken and West’s (1991) suggestion for determining a cut point, which involves a value near the mean plus one standard deviation.

• Aiken, L. S. (1991). *Multiple regression: Testing and interpreting interactions*. Sage Publications

google schola, 2, 513-531.

Response 8: Thank you for your providing important information. We have added that “This value of 3 CPCEs is also consistent with the Aiken and West[37] method recommending that the cut-off value is close to the mean plus one standard deviation ($\text{mean}1.25 + \text{sd}1.4 = 2.65$)” (page 13, lines 5-7). According to the above method, the cut-off point for ACEs would be 2.22 ($\text{mean}0.82 + \text{sd}1.4 = 2.22$), but ACEs=1 was used in this study. This is ‘because of their effectiveness in determining poor health, following those of previous studies[22,30] and the clarity of the semantic difference between “without ACEs” and “with ACEs”’ (page 8, lines 15-17).

Comment 9: Limitations: The researchers should also include the cross-sectional nature of the data, removing the ability to establish causality more specifically adding a citation or citations would be useful.

Response 9: We agree with you. We have added the limitation that “this study could not allow for the determination of causality” (page 17, lines 18-19).

Comment 10: The aforementioned theoretical and methodological concerns must be addressed, even if not all of them can be completely fixed in this one study.

Response 10: We agree with you. We have addressed the above concerns to the best of our ability. Please review the revised draft.

Again, we appreciate all of your insightful comments. We worked hard to be responsive to them. Thank you for taking the time and energy to help us improve the paper.

Reviewer #2

Dr. Tre D. Gissandaner
Columbia University Irving Medical Center

Dear Dr. Tre D. Gissandaner,

Thank you for your enlightening comments. Below, we have provided responses to your comments.

Responses to comments of Reviewer #2:

Comment 1: [Introduction]

1. It would be helpful to the reader to briefly define what a dose-response relationship means. Then a follow-up sentence explaining the implication of this on physical and mental health would provide better justification for the last sentence of this paragraph.

Response 1: We agree with you. We have added that “the greater the exposure to ACEs, the greater the risk of physical and mental illness in later life” (page 4, lines 6-7) as the definition of the dose-response relationship.

Comment 2: [Introduction]

2. Second paragraph. Again, defining or clarifying what the moderating effect of PCEs means for the reader would be helpful.

Response 2: Thank you for your comments. Per your suggestion, we have added that “a moderating effect as a protective factor; that is, the harmful effects of ACEs can be smaller among PCEs-rich individuals than among PCEs-poor individuals” (page 4, lines 9-11).

Comment 3: [Introduction]

3. Paragraph 4. What is meant by “children exposed to ACEs often have ACE in their relationship with caregivers, not PCE”?

Response 3: Your confusion is understandable. The above wording was difficult to understand, so it was replaced with the following: “Children exposed to ACEs were more likely to have been unable to talk to their parents about their feelings or feel safe. In other words, they are less likely to have experienced PCEs in the home” (page 5, lines 3-5).

Comment 4: [Introduction]

4. It would be more appropriate to say that compensatory factors directly impact an outcome independent of the risk factor rather than “neutralize” it. A risk factor could still have a significant effect on the outcome.

Response 4: Thank you for your comments. Per your suggestion, we have dropped the word “neutralize” and expressed “promotional/compensatory factors directly impact developmental outcomes independent of risk factors” (page 6, lines 1-2).

Comment 5: [Methods]

1. Please include how covariate information was obtained and how they were coded for the analyses.

Response 5: Thank you for your comments. Per your suggestion, we have included information on how to answer and code for age, gender, and educational attainment (page 10, lines 2-5).

Comment 6: [Methods]

2. How was relative risk derived?

Response 6: The adjusted predicted prevalence for the adequately exposed group (≥ 3 CPCEs)/adjusted predicted prevalence for the insufficiently exposed group (0-2 CPCEs) $\times 100$ was considered as the relative risk. In the text, this formula was added using ischemic heart disease as an example (page 14, line 19).

Comment 7: [Methods]

3. Was there missing data? Please include how missing data was handled?

Response 7: Thank you for your comments. Per your suggestion, we have added that “due to the nature of the online survey, there were no missing values” (page 7, lines 16-17).

Comment 8: [Results]

1. Please avoid using causal language for this cross-sectional study. Instead of saying “increased,” “decreased,” or “reduced” say something like “was associated with increased odds” or “was associated with lower odds.”

Response 8: We agree with you. The above wording was replaced by the phrase “was associated with lower odds” (page 12, line 9).

Comment 9: [Discussion]

1. Same as above for the discussion section. Please be careful with the causal language.

Response 9: We agree with you. In the discussion section, they were also replaced by the phrase “was associated with increased/lower odds” (page 15, line 3, 16).

Comment 10: [Discussion]

1. The study’s generalizability outside of Japan should be discussed.

Response 10: We agree with you. We have included the following: “The role of CPCEs as a promoting and protective factor found in this study may have been discovered only in the context of Japan, where many local communities have self-governing organizations and traditional events, and violence among residents and students is relatively rare. Future studies should examine whether the findings are generalizable to other countries besides Japan” (page 17, lines 2-6).

Comment 11: [Discussion]

2. In the implications section, language stating “should,” “need,” etc. should be softened.

Response 11: We agree with you. The above expressions were changed to “might as well”(page 18, line 10), “are desirable to”(page 18, line 12), “might be a good way to”(page 18, line 21), and “are expected to”(page 19, line 2).

Comment 12: [Discussion]

3. Could there be additional confounding variables that might impact the internal validity of this study? This needs to be acknowledged.

Response 12: We agree with you. We have added a limitation that “confounding variables (including psychological traits like optimism and hometown regional features, which may be linked to adult diseases, ACEs, and CPCEs) might have gone unnoticed” (page 17, lines 21-23).

Thank you once again for your valuable comments and suggestions. We hope that our revised focus will help you feel better about working.

VERSION 2 – REVIEW

REVIEWER	Mueller , Kyle Curtis Research Analyst at Harris County Juvenile Probation Department
REVIEW RETURNED	26-Feb-2024

GENERAL COMMENTS	Below I provide several comments that I feel were addressed and one that was not addressed within this revised and resubmitted manuscript submission. Addressed: This piece explores the association between ACEs, PCEs, and acquiring disease during adulthood among a cross-sectional internet survey sample of Japanese participants aged 18-79. I enjoyed reading the revised manuscript submission and commend the authors on thinking seriously about positive life experiences, something that has been too often neglected in empirical articles in general in favor of focusing strictly on ACEs and not exploring PCEs at all. Addressed: I liked the edits to the Compensatory Model description. I also liked the addition of the exploration of the moderating relationship that PCEs and ACEs had on adult illness. Also, the researchers took the correct approach when exploring the interaction and the marginal effects and contrast approach utilized for complex situations when examining the association to negative health outcomes. This approach allowed the researchers to come to a lack of statistical support for H3. Not Addressed: The researchers should include in the statistical analysis section the missing data analysis provided in more detail below. The researchers should show that the participants included in the analyses are missing at random rather than missing systematically in which multiple imputations would be needed to be utilized. Specifically, the researcher states that 3,370 and (then difference in the 28,630 and 28,617) leaving 13 adults giving inconsistent results of 28,630 individuals, there were only 28,617 included in Table 2. The researchers should have stated if there
--

	were differences in the key independent variables and dependent variables for those adults included in the study and those adults not included in the study. Addressed: The previous theoretical and methodological concerns have been addressed. I wanted to thank the authors for paying close attention to my previous comments. I believe the manuscript in its current state is eligible for publication in BMJ with minor revisions including missing data analysis within the manuscript.
--	---

REVIEWER	Gissandaner , Tre D. Columbia University Irving Medical Center
REVIEW RETURNED	06-Mar-2024

GENERAL COMMENTS	Thank you for the opportunity to review the revisions for this manuscript. The authors were responsive to my feedback, and I believed the manuscript has been improved in certain aspects as a result. There are still some mostly minor concerns related to study rationale, as well as improving clarity of other aspects of the manuscript. Specific comments and suggestions are detailed below, organized by manuscript sections. Introduction  1. First paragraph. More discussion is still needed to justify the last sentence of this paragraph. This deserves greater explanation for a reader who may be unfamiliar with this process. 2. Second paragraph and elsewhere. When describing the moderating effect of PCEs, please use more inclusive language. "PCEs-rich" could be better described as individuals who reported more PCEs, and "PCEs-poor" as individuals who reported fewer PCEs. 3. Paragraph two. For readability, explanation of the moderating effect of PCEs should go at the end of this paragraph. 4. Third paragraph. This first sentence should go at the end of this paragraph. 5. Paragraph four. The second and third sentences are still confusing. Are there citations to justify these statements? 6. Paragraphs seven, eight, and nine. Some examples of (or findings from) the literature supporting each of these resilience models would be helpful for the reader. As it reads, the revisions made are just hypotheses for the current study—which is helpful, but greater theoretical discussion is needed to understand how the authors reached those hypotheses. Results
---

	1. Could the results be organized in a way to correspond more explicitly with each model or hypothesis of resilience tested? This would improve readability. 2. There is still an instance of causal language used in the results. Page 14, line 7. Please revise accordingly. 3. Please frame “an abundance of CPCEs” or “CPCEs abundance” as “greater reporting of CPCEs” or something similar. This would be more inclusive. Discussion 1. A few remaining instances of causal language. Page 17, line 4—maybe say “with associated relative risk reductions...”; Page 18, line 4—same here, could say “with associated relative risk reductions...”; Page 19, line 8—avoid asserting this study “proves” this process. 2. Please frame “an abundance of CPCEs” or “CPCEs abundance” as “greater reporting of CPCEs” or something similar. This would be more inclusive.
--	---

VERSION 2 – AUTHOR RESPONSE

Reviewer #1

Dr. Kyle Curtis Mueller

Research Analyst at Harris County Juvenile Probation Department

Dear Dr. Kyle Curtis Mueller,

Thank you for your enlightening comments. Below, we have provided responses to each of your comments.

Responses to comments of Reviewer #1:

Comment 1: Addressed: This piece explores the association between ACEs, PCEs, and acquiring disease during adulthood among a cross-sectional internet survey sample of Japanese participants aged 18-79. I enjoyed reading the revised manuscript submission and commend the authors on thinking seriously about positive life experiences, something that has been too often neglected in empirical articles in general in favor of focusing strictly on ACEs and not exploring PCEs at all.

Response 1: Thank you for your comments.

Comment 2: Addressed: I liked the edits to the Compensatory Model description. I also liked the addition of the exploration of the moderating relationship that PCEs and ACEs had on adult illness. Also, the researchers took the correct approach when exploring the interaction and the marginal effects and contrast approach utilized for complex situations when examining the association to negative health outcomes. This approach allowed the researchers to come to a lack of statistical support for H3.

Response 2: Thank you for your comments.

Comment 3: Not Addressed: The researchers should include in the statistical analysis section the missing data analysis provided in more detail below. The researchers should show that the participants included in the analyses are missing at random rather than missing systematically in

which multiple imputations would be needed to be utilized. Specifically, the researcher states that 3,370 and (then difference in the 28,630 and 28,617) leaving 13 adults giving inconsistent results of 28,630 individuals, there were only 28,617 included in Table 2. The researchers should have stated if there were differences in the key independent variables and dependent variables for those adults included in the study and those adults not included in the study.

Response 3: We agree with you. Per your suggestion, we have included an explanation as follows: "There were no significant differences in the key variables (ACEs, PCEs, and adult health outcomes) between 28,630 and 28,617 adults" (page 8, lines 10-11).

Thank you once again for your valuable comments and suggestions. We trust that our revised description will meet your approval.

Reviewer #2

Dr. Tre D. Gissandaner

Columbia University Irving Medical Center

Dear Dr. Tre D. Gissandaner,

Thank you for your valuable comments and suggestions. Below, we have provided responses to your comments.

Responses to comments of Reviewer #2:

Comment 1: [Introduction 1] First paragraph. More discussion is still needed to justify the last sentence of this paragraph. This deserves greater explanation for a reader who may be unfamiliar with this process.

Response 1: We agree with you. Accordingly, we have added the following greater explanation: "Chronic childhood stress exposure can lead to dysregulation of stress response systems, including the hypothalamic-pituitary-adrenocortical axis and the sympathetic-adrenomedullary system, resulting in permanent changes in multiple organ systems, particularly the brain[7]. Epigenetic modifications of DNA could play a role in these changes, and the biological disruptions of brain circuits, other organs, and metabolic systems during sensitive development are thought to lead to an increased risk of various chronic diseases in adulthood[8]" (page 4, lines 7 to 13).

Comment 2: [Introduction 2] Second paragraph and elsewhere. When describing the moderating effect of PCEs, please use more inclusive language. "PCEs-rich" could be better described as individuals who reported more PCEs, and "PCEs-poor" as individuals who reported fewer PCEs.

Response 2: We agree with you. Per your suggestion, we have deleted the terms "PCEs-rich" and "PCEs-poor," and wrote "among individuals who report more PCEs than among those who report fewer PCEs" (page 4, lines 20-21).

Comment 3: [Introduction 3] Paragraph two. For readability, explanation of the moderating effect of PCEs should go at the end of this paragraph.

Response 3: We agree with you. Per your suggestion, explanation of the moderating effect of PCEs ("In other words, PCEs...") has been moved to the end of this paragraph (page 4, lines 19 to 21).

Comment 4: [Introduction 4] Third paragraph. This first sentence should go at the end of this paragraph.

Response 4: We agree with you. Per your suggestion, we have moved first sentence ("Little is understood...") to the end of this paragraph (page 5, lines 5-6).

Comment 5: [Introduction 5] Paragraph four. The second and third sentences are still confusing. Are there citations to justify these statements?

Response 5: Thank you for your comments. We have written as follows: "Compared to those with

fewer ACEs, children exposed to a higher number of ACEs are less likely to report having adults in their lives who made them feel safe or met their basic needs, highlighting a deficit in positive childhood experiences (PCEs)[10]" (page 5, lines 8 to 10).

Comment 6: [Introduction 6] Paragraphs seven, eight, and nine. Some examples of (or findings from) the literature supporting each of these resilience models would be helpful for the reader. As it reads, the revisions made are just hypotheses for the current study—which is helpful, but greater theoretical discussion is needed to understand how the authors reached those hypotheses.

Response 6: We agree with you. Accordingly, we have added the following descriptions of the literature supporting each resilience model:

"The Compensatory (Promotive Factors) Model assumes that promotive/compensatory factors directly impact developmental outcomes independent of risk factors. For example, even after controlling for ACEs, PCEs have been reported to predict less post-traumatic syndrome disorder symptoms and stress in pregnant women[24] and less depression, stress, and sleep difficulties in adults[25]. While ACEs could still have a significant effect on the outcome, CPCEs themselves are thought to enhance health in adulthood.

The Protective Factors Model assumes that protective factors mitigate the relationship between risk factors and bad outcomes. Protective factors do not act independently but can interact with risk factors. Previous studies have shown that adults who protect or meet basic needs[10] and high PCE scores[11] were associated with less poor mental health, especially among those with ACEs. It has also been found that in those with ≥ 4 ACEs, having all childhood community resilience assets such as given opportunity, supportive friends, and role model (vs. none) lowered the prevalence of self-rated poor childhood health from 59.8% to 21.3%[26]. In situations where more CPCEs are reported, the negative impact of ACEs on adult health is likely to be smaller.

The Challenge Model assumes that the modest levels of risk factors protect against additional exposures that could render people prone to unfavorable consequences. Moderate levels of ACEs, particularly when accompanied by more protective factors, may promote improved adult health, akin to inoculation[22]. Some studies did not support this model[21], but it has also been reported that PCE score had greater positive effects on adult health (such as body mass index [BMI] and sleep difficulties) among those with < 4 ACEs than among those with ≥ 4 ACEs[25]. Therefore, it can be speculated whether moderate ACEs can lower the likelihood of adult illness among individuals reporting more CPCEs" (page 6, line 8 to page 7, line 7)

Comment 7: [Results 1] Could the results be organized in a way to correspond more explicitly with each model or hypothesis of resilience tested? This would improve readability.

Response 7: We agree with you. Per your suggestion, we have added explanations regarding three hypotheses for each of our results (page 13, lines 7-8; page 15, lines 2 to 4; page 15, line 10).

Comment 8: [Results 2] There is still an instance of causal language used in the results. Page 14, line 7. Please revise accordingly.

Response 8: We agree with you. The expression "each additional CPCEs decreased the adjusted odds of stroke by 15%" was replaced by the phrase "the adjusted odds of stroke were 15% lower for each additional CPCE" (page 13, lines 8-9).

Comment 9: [Results 3] Please frame "an abundance of CPCEs" or "CPCEs abundance" as "greater reporting of CPCEs" or something similar. This would be more inclusive.

Response 9: We agree with you. Accordingly, in the results section, that phrase was replaced by "reporting more CPCEs" (page 7, line 7; page 15, line 9).

Comment 10: [Discussion 1] A few remaining instances of causal language.

Page 17, line 4—maybe say "with associated relative risk reductions...";

Page 18, line 4—same here, could say "with associated relative risk reductions...";

Page 19, line 8—avoid asserting this study “proves” this process.

Response 10: We agree with you. The expression “with relative risk reductions” was replaced by “with associated relative risk reductions” (page 16, lines 2-3), and the expression “the relative risk reduction” was changed to “the associated relative risk reductions” (page 17, lines 2-3). The phrase “proves” was replaced by “suggest” (page 18, line 5).

Comment 11: [Discussion 2] Please frame “an abundance of CPCEs” or “CPCEs abundance” as “greater reporting of CPCEs” or something similar. This would be more inclusive.

Response 11: We agree with you. The above expression was changed to “reporting more CPCEs” (page 17, line 10).

Once again, we appreciate your insightful comments. We have diligently worked to address them. Thank you for investing your time and effort to assist in enhancing our paper.